# A Comparison of Objectively Measured Free-Living Physical Behaviour in Adults with and without Lower Limb Amputation

**DOI:** 10.3390/ijerph20136198

**Published:** 2023-06-21

**Authors:** Sarah Deans, Alison Kirk, Anthony McGarry, David A. Rowe, Philippa M. Dall

**Affiliations:** 1National Centre for Prosthetics and Orthotics, Department of Biomedical Engineering, University of Strathclyde, Glasgow G4 0NS, UK; sarah@deansltd.com (S.D.); anthony.mcgarry@strath.ac.uk (A.M.); 2Physical Activity for Health, School of Psychological Sciences and Health, University of Strathclyde, Glasgow G1 1QE, UK; darowe999@gmail.com; 3Research Centre for Health, Glasgow Caledonian University, Glasgow G4 0BA, UK; philippa.dall@gcu.ac.uk

**Keywords:** measurement, physical activity, sedentary behaviour, lower limb amputation

## Abstract

Objectively monitored free-living physical behaviours of adults with and without lower limb amputation (LLA) were compared. Methods: 57 adults with LLA wore an activPAL3™ for 8 days. A comparison data set (*n* = 57) matched on gender, age and employment status was used. Variables included: time sitting; standing; stepping; sit-to-stand transitions; step count and cadence. Comparisons were made between adults with and without LLA and between gender, level and cause of amputation. Results: Participants with LLA due to trauma versus circulatory causes were less sedentary and more active; however, no difference in physical behaviour was recorded across gender or level of amputation. Participants with LLA spent more time sitting (*p* < 0.001), less time standing and stepping (*p* < 0.001) and had a lower step count (*p* < 0.001). Participants with LLA took more steps in cadence bands less than 100 steps·min^−1^ and fewer steps in cadence bands greater than 100 steps·min^−1^ compared to participants without LLA. Conclusions: People with LLA were less active and more sedentary than people without LLA and participated in less activity at a moderate or higher intensity when matched on age, gender and employment. Interventions are needed to promote active lifestyles in this population.

## 1. Introduction

Individuals with lower limb amputation (LLA) make up a significant proportion (~70%) of the limb absent population [1]. Approximately 150,000 people undergo amputation of a lower limb in the US each year [1]. Transtibial amputations (TTA) and transfemoral amputations (TFA) (below and above the knee, respectively) have different biomechanical mechanisms which influence the individual’s ability to perform daily living activities [2]. Diseases such as diabetes and peripheral arterial disease can result in lower limb amputation. Around 85% of LLA are due to peripheral arterial disease and LLA are more common in older adults (>65 years) [3]. However, LLA can also be a result of other factors such as trauma. In 2017, 57.7 million people were living with limb amputation due to traumatic causes worldwide [4]. LLA is associated with increased health risks and mortality with an increased risk of falls, muscle weakness and overall poorer quality of life. Fortington et al. [5] reviewed the medical files of patients with TTA and TFA to report on the short-term mortality rates following an LLA procedure. The sample consisted of individuals whose amputation was due to vascular disease/diabetes with a mean age of 74.1 years. It was found that 22% of the population died within 30 days, 44% died within one year and 77% died within five years. 

People who have experienced limb loss, especially due to a chronic pathology, are less likely to lead an active lifestyle [6,7,8]. In addition, the consequences of limb loss, such as prosthetics problems, residual limb conditions and phantom pain can reduce physical activity further and increase negative health effects. Participation in physical activity following amputation does not mirror that of pre-amputation levels, and more barriers than motivations exist to adopting or maintaining a physically active lifestyle, including physical issues such as residual limb pain, psychosocial issues including embarrassment and societal issues such as work hours, cost, accessibility and inclusion [9]. 

Only a few studies have explored the benefits of physical activity for people with LLA, with benefits such as improvements in aerobic and muscular fitness parameters [10], gait performance and reductions in falls [8,11] and improved body image, self-worth and self-efficacy [9] being reported. 

In 2018, the World Health Organisation established the goal to reduce physical inactivity in the general population by 15% by 2030, recommending that programs for physical activity and sports must include all segments of the population to reduce inequalities, including people living with disability. Recently, new guidelines on physical activity and sedentary behaviour have been developed, emphasising that any physical activity is better than none and that regularly undertaking both aerobic and muscle-strengthening activities is important [12,13]. Recommendations were presented for specific groups of the population, including people living with disability. Recommendations for people living with disability are based on a review conducted by Public Health England in 2018 [14]. Various benefits of physical activity for adults with disability are documented in the review, including improved quality of life. Based on the evidence, it was recommended that some physical activity is better than nothing, but for substantial health gains, disabled adults should engage in 150 min of physical activity at a moderate to vigorous intensity per week. They should also do challenging strength and balance exercises on at least 2 days per week. No evidence was found that participation in appropriate physical activities increased the risk of injury or harm in adults with disability. However, these reviews and guidelines encompass all adults living with disability. Further knowledge of the characteristics of different groups of people living with disability regarding physical activity, including those with LLA, is needed to tailor guidelines and interventions to support behaviour change. Few studies have explored objective free-living measurement of physical behaviour in people with LLA. Most such studies have focused on exploring the validity or reliability of activity monitors for use in this clinical group and have been conducted within a laboratory setting [15,16]. A few studies have reported on the total number of steps undertaken by people with LLA, with daily average totals ranging between 6790 steps and 4568 steps (transtibial and transfemoral participants, respectively) [17] and more steps being reported on weekdays than on weekends (6158 steps/day versus 4772 steps/day) [18]. There has been little research exploring patterns of physical behaviour (including sedentary behaviour and intensity of physical activity) within a free-living context. 

In addition, there is little research comparing the free-living physical behaviour of people with limb absence with a healthy, matched control group. Nor has research been undertaken which has compared differences in free-living physical behaviour with different types and causes of LLA. 

The aims of this study were therefore to (1) determine the free-living physical behaviours of adults with LLA, (2) explore differences in physical behaviours with different types and causes of LLA and (3) compare physical behaviours of people with LLA to previously collected data for people without LLA, matched by gender, age and employment status. 

## 2. Materials and Methods

### 2.1. Study Design

This study used a cross-sectional design to collect free-living physical behaviour data from people with LLA. These were compared to previously collected data of matched control participants drawn from a larger database.

### 2.2. Equipment

The activPAL3™ is a small, triaxial device (53 × 35 × 7 mm) weighing approximately 15 gm that attaches to the front of the thigh. A memory capacity of 32 MB and a sampling frequency of 20 Hz allows up to 14 days of recording. The activPAL3 includes an accelerometer which provides information on movement and time which can be used to quantify physical behaviour; and which can be used to determine posture and postural changes using the inclination of the thigh (standing is when the thigh is vertical and sitting/lying when the thigh is horizontal). The reliability and validity of the activPAL3 for measuring physical behaviours in older adults and adults with unilateral LLA has been reported [19,20,21,22]. With these authors, there is consensus that the activPAL is a reliable measurement tool in adults with LLA when used in a laboratory setting. The activPAL shows evidence of relative validity, but not absolute validity. Further evaluation is needed to assess whether similar evidence is found in free-living activity and sedentary contexts. The minimum valid bout length to define a new posture was the default setting of 10 s. This time period was deemed appropriate for the sample of people with limb absence because transitioning between postures may take longer than in a non-clinical population. To achieve an intra-class correlation (*ICC*) of 0.8 and 0.9, respectively, between 5 days and 11 days of monitoring were needed for sitting, between 5 days and 10 days of monitoring for standing and between 7 days and 15 days of monitoring for stepping in a study of older adults [23]. Most previous studies have used a monitoring period of around 7 days [24]. 

### 2.3. Participants 

Participants were adults (aged 18 years and over), with unilateral LLA at either transtibial or transfemoral level of absence. Participants had to be routinely wearing and using a prosthesis for free-living activities and be able to understand the requirements of the study. People who used a wheelchair for periods of the day were not excluded from the study but were asked to describe their daily/weekly wheelchair use in a self-report diary. Ethical approval was granted for the study by the University of Strathclyde Ethics Committee and standard data protection procedures were followed. All participants gave written informed consent prior to participation. 

### 2.4. Recruitment

Email contact was made with 14 UK support groups for people living with amputation, and with approximately 40 individuals working with people with LLA with whom the primary investigator had professional connections. UK private prosthetic practices were also contacted. Within each email, a recruitment poster, a cover letter, a participant information sheet and a consent form were included. Fifteen participants had previously participated in an accelerometer validation and reliability study conducted by the authors [20]. Potential recruits contacted the researcher by email or telephone. If they had not yet received a Participant Information Sheet and Consent Form, this was mailed to them electronically or by post for review and completion. Incentives were not offered, but all participants had the opportunity to receive feedback on their activity patterns and sedentary behaviour via a summary sheet after the monitoring period had been completed.

### 2.5. Procedures

Data were collected between December 2015 and June 2016. If interested in participation, an ID number was assigned and the following study documentation was posted to the participant.
ActivPAL3 monitor with four nitrile waterproof sleeves and four Tegaderm™ adhesive patches.Activity monitor information sheet with verbal, visual and written guidance including a link to an online researcher-led demonstration video showing how to attach and remove the monitor on the non-amputated side.Participant activities of daily living diary.Participant demographic questionnaire.Return address, postage paid, padded envelope identifiable by unique participant ID marked on outer.

Each activPAL3 was initialised to collect data for the maximum period of 14 days prior to posting. Participants were asked to wear their activPAL3 monitor continuously for 24 h each day (including overnight and during water-based activities) and for a minimum of 8 consecutive days to secure at least 7 full days of data and allow comparison with other published work. Participants placed the activPAL3 monitor in a nitrile sleeve and attached it to the mid-point of the front of the thigh using the waterproof dressing. Participants were provided with fresh waterproof sleeves and patches to remove the device and reattach if required. Participants were also asked to note the start day of recording, the time they went to bed and got up each day and any time they removed the monitor. The demographic questionnaire collected information on age, gender, employment status, height, weight, limb absence level and side, cause of amputation and wheelchair use. Participants were asked to post all the study documentation and the activPAL3 monitor back to the research team after at least 8 days of wear time. A second monitor was distributed if there were any issues with monitor malfunction or participant illness and the participant consented to a second delivery. 

### 2.6. Secondary Data Collection—Comparison Group Participants without LLA

Following primary data collection, suitable matched control participants were identified from data previously collected from adults without limb loss. The comparison group from which matched participants were identified was drawn from two sources. (1) a study of 121 adults, with data collected between October 2007 and October 2009 [25] or (2) the control group of 34 adults matched (age, gender, home location) to individuals with Intermittent Claudication, with data collected between June 2013 and December 2013 [26]. Participants were recruited conveniently via staff working at Glasgow Caledonian University for both sources, and also for source two, via posters and leaflets at community locations around Lanarkshire, UK. Participants were aged 18 years and over, community dwelling and self-reported no mobility problems. Ethical approval was provided by the School of Health and Social Care Research Ethics Committee, Glasgow Caledonian University (source 1) and the NHS West of Scotland Research Ethics Committee (source 2). All participants provided written informed consent. Participants were asked to wear a uni-axial activPAL, which recorded acceleration at 10 Hz. The uni-axial activPAL is an earlier version of the tri-axial activPAL3 used in the primary data collection, using the same activity classification algorithm (VANE), and which has been shown to have high levels (>90%) of agreement with the activPAL3 in adults [21,22]. The monitor was programmed to start immediately and set to record for 14 days. The monitor was either not waterproofed (source 1) or waterproofed using nitrile sleeves and waterproof dressing (source 2) and was attached to the front of the thigh using double-sided hypoallergenic adhesive pads (PAL Stickies, PAL Technologies, Glasgow, UK) with an additional waterproof dressing over the top for source 2. Participants were instructed to start wearing the monitor before midnight on the day distributed, to wear continuously including overnight (but to remove when showering, bathing or swimming for source 1), and to stop wearing on the eighth day. For source 1, participants were provided with additional adhesive pads so they could change them each time the monitor was removed. Basic demographic information (age, gender, self-reported height, self-reported weight, employment status and postcode) was reported, but participants were not asked to complete a sleep diary or log wear time.

### 2.7. Matching Procedure

Following primary data collection, suitable matched control participants were identified from the control group dataset. Participants with LLA were matched to the control participants based on three characteristics sequentially; first, matching participants on gender (exact match; male or female), then on age (+/−5 years) and finally employment status (exact match; employed full-time, employed part-time, unemployed or retired). Exact matching on all variables could not be achieved due to the number of matching variables and the size of the database from which the comparison group was drawn. Priority was given to matching on gender and age because these variables were most likely to influence physical activity.

### 2.8. Primary and Secondary Data Processing (All Data)

Data collected and stored on the activPAL3 monitors were downloaded via a USB interface using the activPAL3 proprietary software (activPAL3™ Version 7.2.32) using the VANE algorithm and exported as summary data (presented by week, day and hour) to a Microsoft Excel comma separated values file format (Microsoft Corporation, Redmond, WA, USA). The following variables were produced:Start time and stop time (date and hours and minutes);Total number of steps;Time spent sitting/lying, standing and stepping (decimal proportion of one-hour increments);Sit-to-stand transitions;Energy expenditure (METs/hour);Total steps within 24 cadence bands in increments of 10 steps·min^−1^ (i.e., 1–10 steps·min^−1^, 10–20 steps·min^−1^, … up to > 240 steps·min^−1^).

Outcome measures were calculated for the waking day using the hourly summary data exported from the activPAL3. Outcome measures were summed for each complete hour awake, with data from partial hours at the start and end of the waking day included for all outcome measures except time spent sitting/lying. For each participant, the average daily total was calculated for all outcome measures by dividing the weekly totals by the number of valid days of activity identified. 

Data analysis comparing the physical behaviour of people with and without LLA was performed using a fixed waking period protocol of 16 h. 

For consistency, comparison data were handled in the same way as the data from participants with LLA. The duration of the waking period applied to the comparison group datasets had to match that of the datasets from people with LLA, but the actual waking and sleeping hours did not. A fixed 16-h waking time period was derived from all datasets from people with LLA (*N* = 57) and from a random sample of comparison datasets (*n* = 10). This fixed waking time was calculated by taking the mean waking time in the comparison group and people with LLA and through inspection of the diary data from people with LLA. Identical fixed hours from 07:00 until 23:00 were then applied to both datasets.

### 2.9. Statistical Analysis

Following data download and following examination of each participant’s summary file, any lost, unusable data or data from participants who had not been following instructions were removed from the dataset. All statistical analyses were conducted using IBM SPSS Version 24 (IBM Corp., Armonk, NY, USA). Normal distribution, including equality of variance was checked with the Kolmogorov–Smirnov test using a significance level *p* > 0.05. All variables were distributed normally.

Descriptive statistics were reported for the demographic data of both groups. For people with LLA, descriptive statistics were reported for the physical behaviour outcomes (time spent sitting/lying, standing and stepping; daily step count; sit-to-stand transitions; and energy expenditure). Independent sample *t*-tests were used to compare: (a) differences in physical behaviour outcomes between participants grouped by gender, level of limb absence and cause of limb absence, and (b) differences in demographic characteristics and physical behaviour outcomes between the people with and without LLA. The activPAL3 software exports cadence in hourly time bands and is classified as steps taken in certain cadence bands within each 1-h time period. Independent sample *t*-tests were used to compare each cadence band between the people with and without LLA.

## 3. Results

### 3.1. Participants

Five participants were not included in the final data analysis due to not following the study protocol (*n* = 2), an issue with activPAL3 attachment (*n* = 1) and an insufficient number of days monitored (*n* = 2). From the starting sample size of 62 participants and following the removal of datasets that did not meet the criteria, the number of datasets with 7 full days of useable data available for data reduction and analysis was *N* = 57. Table 1 documents the characteristics of participants with LLA. The home countries of residence of the participants were: Scotland (*n* = 32), England (*n* = 23) and Wales (*n* = 2). 

All LLA participants (*N =* 57) were successfully matched to a participant without LLA based on gender and age. The mean age matching tolerance was 0.58 years (mean difference between participants with and without LLA). Most participants were also matched based on employment status (*n* = 44, 77%). Those who could not be matched perfectly on employment (*n* = 13) were matched as follows: Full-time–Retired, *n* = 7; Part-time–Retired, *n* = 3; Full-time–Part-time, *n* = 2; Full-time–Unemployed, *n* = 1 (employment status of individual with LLA–employment status of control group participant). 

Height, weight and body mass index values were supplied with the comparison group of participants without LLA. Table 2 documents the characteristics of this group. All participants without LLA were residents of Scotland at the time of testing. With the exception of body mass, where the control group were lighter than the individuals with LLA (*t*_(56)_ = −2.05, *p* = 0.045, *d* = 0.4), there were no significant differences in the other sample characteristics of age, height and body mass index.

### 3.2. Physical Behaviour of Participants with Lower Limb Absence

Table 3 presents the average daily physical behaviour for participants with LLA (*N* = 57) by gender during waking time including: time spent awake, asleep, sitting/lying, standing and stepping; average daily step count; sit-to-stand transitions; and energy expenditure. Actual values expressed in terms of the percentage of the waking day are also presented. Independent *t*-tests identified no significant differences in any of these variables by gender. Table 4 presents the average daily values by level of limb absence for all physical behaviour variables. Independent *t*-tests identified no significant differences in any variable by level of limb absence. In considering the cause of limb absence, only one participant had limb absence due to thrombosis. Therefore, the causes peripheral arterial disease and thrombosis were combined to create the group titled Circulatory for meaningful differences to be tested using an independent *t*-test.

There were no significant differences between any of the physical behaviour variables for trauma and the groupings of cancer, infection and congenital absence (Table 5). However, in comparing trauma and circulatory groupings, there were statistically significant (*p* < 0.05) differences in all physical behaviours except for sit-to-stand transitions (*p* = 0.82). Those who had sustained amputation due to circulatory causes spent more time sitting/lying throughout the day and less time standing and stepping than those people who had amputation due to trauma. In addition, those in the trauma grouping took more daily steps and performed more sit-to-stand transitions than those in the circulatory grouping. Finally, those with circulatory issues had lower daily energy expenditure than those who had experienced traumatic amputation.

The average daily number of steps achieved by people with LLA within cadence bands is shown in Figure 1. The greatest number of steps recorded was within the three cadence bands of 80–90, 90–100, and 100–110 steps/min. In all cadence bands of 100 steps/minute and above (equivalent to moderate-to-vigorous physical activity), a daily average of 1889 steps was recorded.

### 3.3. Physical Behaviour of Participants with and without LLA 

Table 6 shows the physical behaviour variables for participants with and without LLA. Independent sample *t*-tests were performed to compare the physical behaviour of the two groups. All physical behaviour variables were significantly (*p* < 0.05) different between participants with and without LLA, with the exception of the number of daily sit-to-stand transitions, in spite of a medium effect size (*t*_(56)_ = 1.80, *p* = 0.107, *d* = 0.39). The number of sit-to-stand transitions was 12% greater for people with LLA than for those without LLA despite the daily standing and stepping time being lower. Except for the sit-to-stand variable, physical behaviour outcomes were more favourable for participants without LLA compared to people with LLA. As an example, participants without LLA took 39% more daily steps than participants with LLA.

Step cadence was compared between the two participant groups (Figure 1). Participants without LLA took more steps across all cadence bands except for the 70–80 steps/minute cadence band. Of particular note is the comparison in average daily cadence band values in the three cadence band ranges spanning 100–130 steps/minute where participants without LLA achieved more daily steps (*p* < 0.01) for all three cadence bands. For all the cadence bands of 100 steps/minute and above, participants without LLA took 4657 average daily steps compared to participants with LLA who took an average of 1889 steps (*t*_(56)_ = −7.73, *p* < 0.01, *d* = 1.76). Independent samples *t*-tests showed that there were significant differences in all cadence bands of 100 steps/minute and above, with the exception of two cadence bands of 140–150 steps/minute (*t*_(56)_ = −1.94, *p* = 0.058, *d* = 0.76) and 170–180 steps/minute (*t*_(56)_ = −1.84, *p* = 0.071, *d* = 0.92). When step cadence is expressed as a percentage of the total steps taken (Figure 2), participants with LLA took a greater proportion of daily steps in all step cadence bands less than 100 steps/minute but a smaller proportion of daily steps in all step cadence bands greater than 100 steps/minute. 

## 4. Discussion

We examined the free-living physical behaviour of adults with LLA and compared this to adults without LLA matched on gender, age and occupation. People with LLA participate in less physical activity and more sedentary behaviour than those who do not have limb absence. The sample was representative of the national limb absent population in terms of gender, with the proportion of male and female recruits being 70.2% and 29.8%, respectively. The gender split for the UK national limb absent population during 2011–2012 was 69.9% and 30.2%, respectively [27]. The relative proportion of study participants with transtibial limb absence in this study was higher than the national average (69.9% versus 59.3%) and the relative proportion with transfemoral absence was lower than the national average (30.1% versus 40.7%). Almost half of the participants with limb absence (49.1%) sustained amputation due to trauma. This could be regarded as being unrepresentative of the general UK population with amputation/limb absence where trauma accounts for only 18.5% of amputations [27]. This may have influenced the findings of this study, as differences have been reported between those who have limb absence due to peripheral arterial disease and those who had experienced limb loss due to trauma, with the latter reporting higher levels of physical activity. This was also what we found in this study. Physical activity after dysvascular lower limb amputation is influenced by relationships among health understanding, motivation, support and self-efficacy in the presence of disability [28].

### 4.1. Physical Behaviour of Participants with Lower Limb Absence

The average daily steps for participants with LLA was 5569 ± 4083. The average percentage of time spent sitting, standing and stepping was: 70.5%; 21.2% and 8.3% with an average of 57 daily sit-to-stand transitions. There was no difference in physical behaviour across gender or level of amputation in the LLA group. Participants with LLA due to trauma spent less time sitting and more time standing and stepping, with a higher step count and daily sit-to-stand transitions compared to participants with limb absence due to circulatory causes. This finding is comparable to previously published work. Stepien et al., 2007 [17] conducted a study to examine activity levels using a StepWatch3 Activity Monitor in 77 adults with LLA. The StepWatch3 Activity Monitor was fitted to each participant’s prosthesis and was programmed to record 6 days of complete data. Stepien et al. (2007) [17] reported that participants averaged 6126 steps per day. The authors also reported that those with transtibial absence (*n* = 54) took, on average, 6790 steps/day, and those with transfemoral absence (*n* = 23) took, on average, 4568 steps per day. 

### 4.2. Comparison of Participants with and without Lower Limb Absence 

Participants with LLA spent more time sitting and less time standing and stepping and had a lower daily step count but a higher number of daily sit-to-stand transitions in comparison to participants without LLA. In comparison to participants without LLA, participants with LLA took fewer steps within each cadence band; however, when steps were expressed as a percentage of total steps taken, participants with LLA took a greater proportion of daily steps in cadence bands less than 100 steps/minute and a lower proportion of daily steps in cadence bands greater than 100 steps/minute. 

Moderate-intensity physical activity is defined as 3–6 METs. Under laboratory testing conditions in two mixed-gender studies with healthy participants (*n* = 75 and *n* = 50 respectively), cadence at 3 Metabolic Equivalents (METs) has been established as 103 steps/minute and 102 steps/minute, respectively [29,30]. A systematic review has shown that a proportion of steps taken at the rate of 100 steps/minute can be prescribed for adults in order to achieve this activity intensity [31,32]. Another study examined cadence during music-prompted and self-regulated walking in adults with LLA [16]. In this study of 17 participants (15 men, 2 women) with unilateral transtibial amputation, it was reported that walking briskly at 86 steps/minute corresponded to an intensity of 3 METs. This suggests that the participants with LLA in this study may be achieving a comparable number of steps at moderate intensity levels and above, albeit at a lower step cadence, although this requires further exploration. It has also been reported previously that cadence may differ based on the level and cause of amputation [33] and this also needs further exploration. 

The concept of interrupting sedentary behaviour with standing and stepping behaviours has emerged as a potential way of modifying the detrimental effects on health caused by sedentary behaviour [34,35]. Bohannon et al. (2015) [36] published a review of sit-to-stand transitions as a component of everyday mobility in both healthy people and those with diagnosed conditions such as stroke, cancer and osteoarthritis. The mean number of daily transitions was at least 45 for all groups except for a group of community-dwelling older adults where 39 transitions were recorded. In comparing the daily transitions recorded for the participants in this study with those of other clinical populations, the data compare favourably with participants with lower limb absence achieving between 54 and 57 transitions daily (individualised and fixed waking period values, respectively).

### 4.3. Strengths, Limitations and Future Work

The key strength of this study was the determination of objectively measured free-living physical behaviour of people with LLA compared to a matched control sample without mobility problems. Another strength of the study was the recruitment methodology, leading to a high participation rate. Guidance on protocols to collect quality objective physical activity and sedentary behaviour data have been published and a number of these stringent methodology principles were followed in this study [37]. Data from only five participants were unsuitable for inclusion in the data analysis as compliance with study protocols was high. The number of valid datasets was also optimised by re-testing participants if their initial dataset did not yield suitable data; three participants undertook re-testing. The average number of full days of recorded data achieved was 7.7 days which is higher than the number of full days achieved in other comparable studies which have used the activPAL3 accelerometer [38].

The study also contained limitations. The possibility of behaviour modification simply as a result of the novelty of participating in a research study and being observed must be considered and taken into account in the interpretation of the results. Secondly, there are limitations in the matching of the study group with the control group including differences in: the model of activPAL used (one being uniaxial and one triaxial); the year and month of data collection and potentially characteristics of age, gender and employment status which could have had an impact on the physical activity and sedentary behaviour recorded. The findings related to the comparison of physical activity and sedentary behaviour between people with and without LLA therefore remain indicative and require further exploration. Finally, some of the study participants stated in their daily self-reporting diaries that they used a wheelchair for part or all of the day. We felt it was important to include them as representative of the study population. However, information was not collected on the time of wheelchair use and therefore data collected during wheelchair use was not excluded or analysed differently. This should be considered in future research.

In future work, the generalisability of the study could be extended with larger sample sizes, which could allow further comparison across levels and causes of amputation to be further explored. A more representative sample in terms of cause of amputation could be achieved by recruiting from healthcare settings. Further examination of physical activity and sedentary bout duration and patterns of physical behaviour during week days and weekend days could also be explored. Differences in sit-to-stand transitions should also be explored. This research showed that despite being more sedentary and less active, the LLA sample engaged in more sit-stand transitions. Despite longer periods of sitting, there is the implication that the LLA population may have more healthy sedentary patterns if sitting is interrupted more frequently. This warrants further work.

## 5. Conclusions

This study demonstrates that people with LLA participate in less physical activity and more sedentary behaviour than people without LLA when matched on age, gender and employment. Participants with LLA due to trauma spent less time sitting and more time standing and stepping, with a higher step count and daily sit-to-stand transitions, compared to participants with limb absence due to circulatory causes. Findings support the need for the development and implementation of clinical and community-based interventions to support an active lifestyle for individuals with LLA, particularly when related to circulatory causes.

## Figures and Tables

**Figure 1 ijerph-20-06198-f001:**
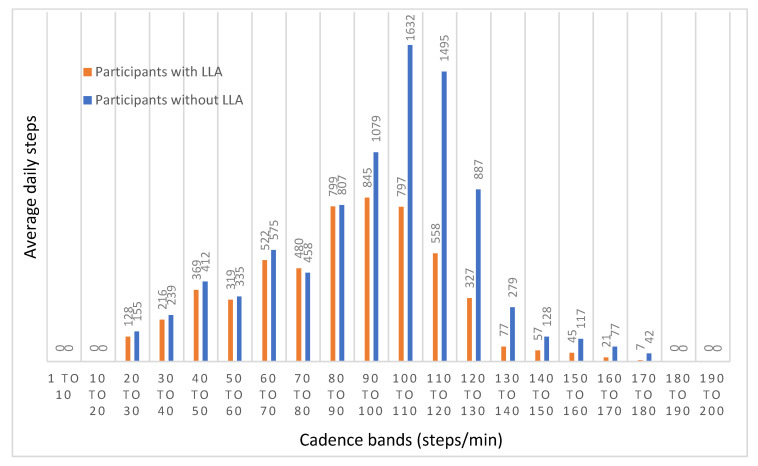
Average daily steps within each cadence band for people with lower limb absence compared to comparison group.

**Figure 2 ijerph-20-06198-f002:**
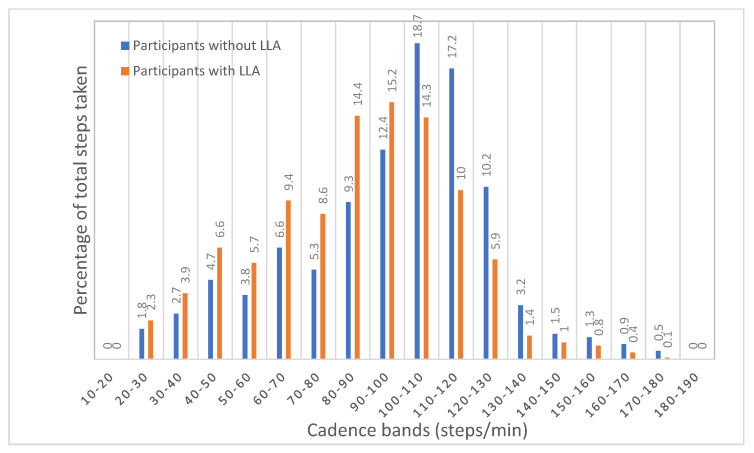
Comparison of step cadence, expressed as a percentage of total steps for participants with lower limb absence compared to comparison group.

**Table 1 ijerph-20-06198-t001:** Characteristics of participants with lower limb absence.

Characteristic		All (*N* = 57)	Male (*n* = 40)	Female (*n* = 17)
Age (years) ^1^		57.39 ± 12.09	59.40 ± 10.85	52.65 ± 13.82
Height (m) ^1^		1.73 ± 0.10	1.77 ± 0.07	1.63 ± 0.07
Body mass (kg) ^1^		81.48 ± 16.99	87.58 ± 14.67	67.14 ± 13.17
Body mass index (kg/m^2^) ^1^		27.14 ± 4.54	27.94 ± 4.26	25.27 ± 4.76
Level	Transtibial	40	27	13
	Transfemoral	17	13	4
Side	Right	29	22	7
	Left	28	18	10
Cause of limb absence	Trauma	28	22	6
	Cancer	8	3	5
	Infection	8	6	2
	Congenital	5	3	2
	PAD	7	5	2
	Thrombosis	1	1	0
Employment status	Retired	28	22	6
	Full-time	18	13	5
	Part-time	9	4	5
	Unemployed	2	1	1
Wheelchair user	No	40	31	9
	Yes	17	9	8

Note. ^1^ values are means ± 1 standard deviation, PAD = Peripheral Arterial Disease.

**Table 2 ijerph-20-06198-t002:** Characteristics of comparison group participants without lower limb absence.

Measurement	All (*N* = 57)	Male (*n* = 40)	Female (*n* = 17)
Age (years) ^1^	57.86 ± 14.10	59.92 ± 13.48	53.00 ± 14.73
Height (metres) ^1^	1.72 ± 0.10	1.77 ± 0.07	1.61 ± 0.06
Body mass (kilograms) ^1^	75.94 ± 10.99	79.61 ± 10.14	67.30 ± 7.69
Body mass index (kg/m^2^) ^1^	25.74 ± 3.22	25.80 ± 3.42	25.58 ± 2.79

Note. ^1^ values are means ± 1 standard deviation.

**Table 3 ijerph-20-06198-t003:** Average daily physical behaviour of participants with lower limb absence by gender.

Variable	All (*N* = 57)	Male (*n* = 40)	Female (*n* = 17)
Time awake (h)	14.88 ± 0.98	14.98 ± 0.98	14.66 ± 0.95
Time awake (%)	62.02 ± 4.07	62.42 ± 4.08	61.07 ± 3.98
Time asleep (h)	9.12 ± 0.98	9.02 ± 0.98	9.34 ± 0.95
Time asleep (%)	37.98 ± 4.07	37.57 ± 4.09	38.93 ± 3.98
Time sitting/lying (h)	10.44 ± 10.99	10.61 ± 1.91	10.03 ± 1.58
Time sitting/lying (%)	70.45 ± 12.92	71.35 ± 14.28	68.35 ± 8.98
Time standing (h)	3.19 ± 1.43	3.10 ± 1.58	3.42 ± 1.01
Time standing (%)	21.24 ± 8.91	20.37 ± 9.66	23.30 ± 6.65
Time stepping (h)	1.25 ± 0.82	1.27 ± 0.91	1.21 ± 0.59
Time stepping (%)	8.31 ± 5.13	8.29 ± 5.56	8.35 ± 4.11
Daily step count	5569 ± 4083	5677 ± 4535	5316 ± 2852
Number of daily sit-to-stand transitions	57 ± 20.00	54 ± 20.00	65 ± 20.00
Daily energy expenditure (METs/h) ^1^	7.32 ± 2.39	7.46 ± 2.70	6.97 ± 1.45

Note. ^1^ Adjusted to remove basal waking hours energy expenditure. Values are means ± 1 standard deviation.

**Table 4 ijerph-20-06198-t004:** Average individualised daily physical behaviour of participants by level of limb absence.

Activity Variable	Transtibial (*n* = 40)	Transfemoral (*n* = 17)
Time awake (h)	14.91 ± 0.98	14.82 ± 0.99
Time awake (%)	62.12 ± 4.10	61.77 ± 4.12
Time asleep (h)	9.09 ± 0.98	9.17 ± 0.99
Time asleep (%)	37.87 ± 4.09	38.23 ± 4.12
Time sitting/lying (h)	10.25 ± 1.92	10.90 ± 1.55
Time sitting/lying (%)	69.06 ± 13.60	73.72 ± 10.83
Time standing (h)	3.32 ± 1.49	2.89 ± 1.29
Time standing (%)	22.06 ± 9.21	19.32 ± 8.11
Time stepping (h)	1.34 ± 0.90	1.04 ± 0.58
Time stepping (%)	8.88 ± 5.56	6.95 ± 3.74
Daily step count	6085 ± 4449	4356 ± 2809
Number of daily sit-to-stand transitions	60 ± 21	50 ± 15
Daily energy expenditure (METs/h) ^1^	7.56 ± 2.56	6.75 ± 1.89

Note. ^1^ Adjusted to remove base waking hours energy expenditure. Values are means ± 1 standard deviation.

**Table 5 ijerph-20-06198-t005:** Average individualised daily physical behaviour by cause of lower limb absence.

Variable	Trauma (*n* = 28)	Cancer (*n* = 8)	Infection (*n* = 8)	Congenital (*n* = 5)	Circulatory (*n* = 8) ^1^
Time awake (h)	14.84 ± 0.97	15.47 ± 0.76	14.70 ± 0.84	14.65 ± 1.54	14.78 ± 0.96
Time awake (%)	61.84 ± 4.03	64.47 ± 3.15	61.25 ± 3.48	61.05 ± 6.41	61.57 ± 3.98
Time asleep (h)	9.16 ± 0.97	8.53 ± 0.76	9.30 ± 0.84	9.35 ± 1.54	9.22 ± 0.96
Time asleep (%)	38.16 ± 4.06	35.53 ± 3.15	38.75 ± 3.48	38.95 ± 6.41	38.43 ± 3.98
Time sitting/lying (h)	9.75 ± 1.74	10.59 ± 1.95	10.99 ± 1.46	10.59 ± 1.61	12.07 ± 1.48
Time sitting/lying (%)	66.07 ± 12.87	68.63 ± 13.15	74.64 ± 7.53	72.87 ± 13.13	81.89 ± 10.71
Time standing (h)	3.57 ± 1.37	3.47 ± 1.47	2.65 ± 0.74	2.94 ± 1.96	2.28 ± 1.52
Time standing (%)	23.88 ± 8.39	22.28 ± 9.01	18.17 ± 5.51	19.35 ± 11.32	15.24 ± 9.79
Time stepping (h)	1.52 ± 0.92	1.41 ± 0.73	1.05 ± 0.50	1.12 ± 0.50	0.42 ± 0.21
Time stepping (%)	10.05 ± 5.56	9.09 ± 4.60	7.19 ± 3.44	7.78 ± 3.84	2.87 ± 1.45
Daily step count	6817 ±4607	6376 ±3583	4536 ±2552	5449 ±2790	1502 ± 778
Number of daily sit-to-stand transitions	61 ± 20	58 ± 20	54 ± 13	61 ± 26	46 ± 22
Daily energy expenditure (METs/h) ^2^	7.72 ± 2.80	8.51 ± 1.87	6.64 ± 1.47	7.02 ± 1.89	5.59 ± 1.26

Note. ^1^ Circulatory group comprises the conditions peripheral arterial disease and thrombosis combined ^2^ Adjusted to remove base waking hours energy expenditure. Values are means ± 1 standard deviation.

**Table 6 ijerph-20-06198-t006:** Average daily physical behaviour participants with and without LLA.

Variable	Statistic	LLA *N* = 57	CG *N* = 57	Percentage Difference % LLA vs. CG ^2^	*df*	*t*	*p*
Time sitting/lying (h)	*M*	11.73	10.41	13	56	−3.42	<0.001 *
	*SD*	1.99	1.38				
Time standing (h)	*M*	3.07	3.98	23	56	2.29	<0.001 *
	*SD*	1.38	1.11				
Time stepping (h)	*M*	1.20	1.75	31	56	3.31	<0.001 *
	*SD*	0.75	0.49				
Step count	*M*	5340	8715	39	56	3.22	<0.001 *
	*SD*	3613	2639				
Sit-to-stand transitions	*M*	54	48	12	56	1.80	0.107
	*SD*	20	11				
Energy expenditure (MET/h) ^1^	*M*	22.53	24.13	20	56	2.08	<0.001 *
	*SD*	1.66	1.21				

Note. All variances were tested with Levene tests and found to be equal, * denotes significant difference, ^1^ Adjusted to remove base waking hours energy expenditure, ^2^ Calculated as a percentage of the CG result.

## Data Availability

All data are stored on the secure University of Strathclyde OneDrive system. These data can be made available by contacting the lead author.

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
