# Peer review of "A Comparison of Objectively Measured Free-Living Physical Behaviour in Adults with and without Lower Limb Amputation"

_ijerph, 2023, doi:10.3390/ijerph20136198_

Round 1

Reviewer 1 Report

General questions

The manuscript has merit because it has interesting and relevant results about locomotor parameters of people with lower limb amputations. It has adequate theoretical justification, design, statistical analysis and discussion. However, some items need to be corrected and clarified, especially in the discussion of the results.

 Major questions

- Table 3: Please, review time awake % (all): 2.02 (its not correct)

- despite being described in the text, some type of simple sign to indicate a significant difference could be inserted in the tables, in order to facilitate the analysis of the results.

- Linha 324: I think the results failed to highlight that patients without amputation had higher average daily steps at a cadence of 100-110 steps/min and patients with amputation had a greater number of daily steps at a cadence of 90-100 steps/min. This means that the amputation not only reduced the average and number of daily steps, but also the frequency of the most adopted strides throughout the day. In addition, this result indirectly suggests a lower average walking speed throughout the day (if the theoretical assumption that an amputee walks with a shorter stride length).

- Figure 2: I think it is important to highlight that in the highest stride frequencies, patients with amputation had a higher percentage of daily steps compared to individuals without amputation. This interpretation is not contradictory with the previous interpretation, since it could be assumed that the stride length is shorter. The cadence of the highest percentage of daily steps in amputees was 100-110 and non-amputee 90-100. This result seems to be a consequence of a shorter stride length possibly associated with the need to correct balance by increasing stride frequency.

- Line 392: “In comparison to participants without LLA, participants with LLA took fewer steps within each cadence band,”. This statement does not seem correct to me, see figure 1

- Line 405: “This suggests that the participants with LLA in this study may not be achieving enough steps at moderate intensity levels and above, although this requires further exploration”. This sentence does not match the information on line 403-405. If in the aforementioned study, patients with LLA achieved 3 METs with a stride frequency of 80/min, then they are reaching a greater metabolic power with a lower stride frequency. That is, the activity is being challenging for them with less stride frequency. This makes sense if we analyze from an energetic point of view with the increase in co-contractions, for example, which generate less mechanical work and increase metabolic energy expenditure and reduce the mechanical efficiency of walking (Peyré-Tartaruga and Coertjens 2018: https://www.frontiersin.org/articles/10.3389/fphys.2018.01789/full ). In addition, this activity must be generating greater energy needs due to cardiorespiratory work. This discussion needs to be related to the comment made above regarding figure 2 about walking speed (that is, they seem to walk more slowly and spend more energy for the same speed)

- Please, improve the discussion about the differences verified in relation to the causes of amputation (trauma versus circulatory causes) (please, also include this result in the conclusion)

- Are the x-axis of figures 1 and 2 the same? It seems to me they are the same. Thus, I see no sense in describing the x-axis differently between the two figures.

Minor questions

- Table 1: PAD: ?

- References: a) title of journals: and or &? Please standardize; b) reference 29: initial letters of the title of the manuscript in capital letters; c) reference 14: italics; d) reference 01: abbreviated and unformatted journal title

Author Response

Please see attached file for details of changes made in response to comments from reviewer 1. 

Reviewer 2 Report

The study's goals were threefold: (1) to determine the physical behavior of adults with ALL, (2) to investigate differences in physical behavior based on different types and causes of ALL, and (3) to compare the body behaviors of people with ALL to data previously collected for people without ALL, based on gender, age, and employment status. This is an intriguing, well-designed, and well-written study, in my opinion. The first two objectives were met rigidly and satisfactorily, however, the third objective is the study's fundamental flaw. In fact, we typically compare data collected under the same settings, with the same instruments, and between two groups that differ only in the variable(s) under investigation (LLA in the current study).  The authors have clearly done their utmost to make the two groups similar, but this is insufficient. The active tri-axisPAL3 was used in the experimental group, whereas the uniaxial activePal3 was used in the control group; the concordance between the two devices is roughly 90%, so a 10% discrepancy is significant and could skew the results. Between the first and second measurements, around 15 years passed, a considerable period during which several changes in living and working situations occurred. Finally, participants with LLA were matched to control participants based on gender, age, and employment status (full-time, part-time, or retired employees); nevertheless, these conditions are insufficient because two full-time employees may engage in fundamentally different physical activities. Nevertheless, this part could remain indicative and the authors will have to clearly indicate this in the methodology, discussion, and limitations of the study.

Minor Comments

Ø  Line 30: Please add a reference.

Ø  Line 72: The authors stated that it was recommended that adults with disabilities aim for at least 150 minutes per week of moderate-intensity activity. I think that is not exactly what was recommended by Public Health England in 2018. Please go back to the document and give the exact recommendation.

Ø  Line 119: Please replace (Reid et 119 al., 2013) with a number in brackets.

Ø  Line 122: Please move this paragraph to the end of the "Participants" section.

Ø  Table 1 (line 9) and Table 3 (line 3): please revise the values.

Ø  Please mention the results of the different comparisons in the corresponding tables, as in Table 6.

Ø  Please check for equality of variance in all comparisons where the numbers are not equal, not just in Table 6.  

Ø  Lines 382 and 412: Please correct the references

Author Response

Please see attached file for details of changes made in response to comments from reviewer 2. 

Round 2

Reviewer 1 Report

General questions

The manuscript has merit because it has interesting and relevant results about locomotor parameters of people with lower limb amputations. It has adequate theoretical justification, design, statistical analysis and discussion. The authors clarified the doubts and corrected the requested questions.